# A webcam-based machine learning approach for three-dimensional range of motion evaluation

Xiaoye Michael Wang[1]*, Derek T. Smith[2], Qin Zhu[2]

1 Faculty of Kinesiology and Physical Education, University of Toronto, Toronto, ON, Canada, 2 Division of Kinesiology and Health, University of Wyoming, Laramie, WY, United States of America

* michaelwxy.wang@utoronto.ca

## Abstract

### Background

Joint range of motion (ROM) is an important quantitative measure for physical therapy. Commonly relying on a goniometer, accurate and reliable ROM measurement requires extensive training and practice. This, in turn, imposes a significant barrier for those who have limited in-person access to healthcare.

### Objective

The current study presents and evaluates an alternative machine learning-based ROM evaluation method that could be remotely accessed via a webcam.

### Methods

To evaluate its reliability, the ROM measurements for a diverse set of joints (neck, spine, and upper and lower extremities) derived using this method were compared to those obtained from a marker-based optical motion capture system.

### Results

Data collected from 25 healthy adults demonstrated that the webcam solution exhibited high test-retest reliability, with substantial to almost perfect intraclass correlation coefficients for most joints. Compared with the marker-based system, the webcam-based system demonstrated substantial to almost perfect inter-rater reliability for some joints, and lower inter-rater reliability for other joints (e.g., shoulder flexion and elbow flexion), which could be attributed to the reduced sensitivity to joint locations at the apex of the movement.

### Conclusions

The proposed webcam-based method exhibited high test-retest and inter-rater reliability, making it a versatile alternative for existing ROM evaluation methods in clinical practice and the tele-implementation of physical therapy and rehabilitation.

**Data Availability Statement:** Data cannot be shared publicly because of ethics and privacy concerns. Data could be made available upon request to Nichole Person (Research Compliance

Coordinator, Office of Research Economic
Development, University of Wyoming) at
njperson@uwyo.edu.

**Funding:** National Institute on Minority Health and
Health Disparities, 1R41MD015689-01, Qin Zhu
US Department of the Treasure and Wyoming
Health & Bioscience Innovation Hub, CARES-HUB-
MOVE#1040, Derek T. Smith The funders had no
role in study design, data collection and analysis,
decision to publish, or preparation of the
manuscript.

**Competing interests:** The authors have declared
that no competing interests exist.

## Introduction

The recent COVID-19 pandemic has accentuated the need of improving access to healthcare
services for vulnerable populations. To improve accessibility, various medical fields are making
greater use of telehealth. Telehealth leverages the convenience of personal electronic devices to
enable the remote delivery of healthcare services, such as through telephone consultation and
videoconferencing. As a result of the booming personal electronic device market, telehealth's
popularity has been consistently increasing over the past decade, especially during the
COVID-19 pandemic [1]. Compared to traditional in-person care, telehealth clinical effective-
ness has been shown to be equivalent or better [2, 3]. Unfortunately, the main bottleneck for
telehealth is devising objective outcome measures that are equally accessible.

In the context of physical therapy and rehabilitation, joint range of motion (ROM) is a
widely used outcome measure. One of the most common ways to measure joint ROM is
through a handheld goniometer [4–8]. ROM measurement using a goniometer requires certi-
fied physicians or physical therapists with significant training and practice in using the device.
This imposes challenges to medically underserved communities (rural and remote communi-
ties), where access to healthcare and trained professionals is limited. Furthermore, although the
goniometer has been widely used for ROM evaluation [9, 10], its precision and inter-tester reli-
ability could be low due to human error and procedural inconsistency [11–13]. As a result,
there is an exigency for an alternative ROM evaluation protocol that could address the accessi-
bility, accuracy, and reliability issues related to the goniometer-based ROM evaluation method.

Various non-traditional solutions for ROM evaluation have been proposed over the years,
such as digital photography [9, 14], photogrammetry [10], and even optical motion capture
(MoCap) [15]. MoCap systems, such as OptiTrack (NaturalPoint, Corvallis, OR, USA) and
Vicon (Oxford metrics, UK), have been widely used in biomechanics and sports science stud-
ies as they demonstrate high accuracy (± 0.10 mm) and reliability at a high sampling frequency
(up to 1000 FPS) [16, 17]. However, despite the advantages that MoCap systems offer, they not
only are expensive but also require technical proficiency and large physical space, which pres-
ents noticeable barriers to entry for broad clinical application. In recent years, the field of
machine learning-based human pose estimation has seen significant development [18–20].
These algorithms, such as convolutional neural networks (CNN), [21] use various deep neural
network architectures to identify, recover, and track the 2D or 3D locations of key joints of a
human actor across multiple frames using a single image stream [22].

Capitalizing on the advances in computer vision, a framework called *MediaPipe* offers free
access to machine learning solutions to various computer vision problems [23], including
image-based real-time three-dimensional (3D) pose estimation. The pose-estimation compo-
nent, called *BlazePose* [24], uses a CNN architecture that combines a lightweight pose detec-
tion network with a pose prediction network. The pose detection network first detects any
person presented in a single frame while the prediction network tracks the person across sub-
sequent frames. This design allows the machine learning model to consistently track 33 body
landmarks in real-time at over 30 frames per second using only a single stream RGB video,
such as through a webcam. Compared to other open-sourced 3D pose estimation libraries,
such as OpenPose [19, 25], BlazePose is computationally lightweight and can be deployed
across various platforms, such as in a web browser as a JavaScript application. As a result of its
versatility, BlazePose and other video-based pose estimation libraries have been used by
researchers to develop different applications, including in the context of sports for movement
abnormality detection [26], gait assessment [27], hypermobility assessment [28], yoga training

[29], postural disorder monitoring for Parkinson's patients, [30] and spinal dysfunction risk estimation [31]. However, studies that explicitly examine the effectiveness of applying open-sourced 3D pose-estimation library for range of motion evaluation remain scarce [32].

The current study presents an alternative ROM evaluation method that leverages the power of computer vision algorithms. Using a single webcam and the output from BlazePose, 3D positions of various joints were recorded and used to estimate their corresponding ROM angles. Results obtained using this lightweight machine learning solution were compared to those produced by a MoCap system (OptiTrack) as the ground truth. Additionally, unlike previous studies [9, 10, 14, 15, 21] that only focus on a single joint, the current study evaluated the reliability of ROM measurements using a full-body multi-joint model. Results showed that the proposed solution not only offers an alternative to the traditional ROM evaluation methods with a noticeably lower barrier of access, but also provides high intra- and inter-rater reliability necessary for practical usage.

## Methods

### Participants

Twenty-five healthy adults (12 males, 13 females) from the University of Wyoming community volunteered to participate between March and November 2022. This study was approved by the University of Wyoming Institutional Review Board (IRB). All participants provided their written informed consent prior to the study.

### Data acquisition

Data acquisition was performed using a desktop computer with an Intel Core i9 11th Gen CPU, Nvidia RTX 3090 graphics card, and 64 GB of RAM. To derive the range of motion (ROM) angles for different joints, joint trajectories of various ROM evaluation movements were simultaneously recorded using a webcam-based, real-time pose estimation algorithm and a marker-based infrared optical motion capture system. A custom Python program was implemented to stream and record trajectory data from both sources on a synchronized time scale.

For the webcam-based solution, the pose estimation component of Google's MediaPipe framework [23] (BlazePose [24]), was used. The video was streamed through a Logitech C922 Pro HD Stream Webcam mounted on a computer monitor, approximately 4 m away from the participants at a height of around their chest. The placement of the webcam was to ensure the participants' entire body, including the vertically or laterally extended arms, was visible and centered in the video frame. The video capture was controlled through the OpenCV Library [33] for Python. Although BlazePose could process data at up to 30 frames per second (FPS), due to constraints imposed by simultaneous data streaming from OptiTrack and OpenCV and Python's global interpreter lock (GIL), the effective framerate for BlazePose was at around 15 FPS. **Fig 1** (left) shows the 33 joints obtained from the algorithm with selected labels. BlazePose can generate 3D coordinates of the 33 joints in the world coordinate system (the depth axis represents the relative distance to the camera), along with an estimated "visibility" index for each joint at each frame. The visibility index (ranges between 0 and 1) tracks the algorithm's confidence in the derived joint coordinates, which could be affected by factors such as occlusion and movement speed. A low visibility index indicates inaccuracy in the derived coordinates. A threshold for visibility was set at 0.5 and any trajectory points with an index below the threshold were removed from the analysis. Overall, there were only four trials (or 0.13% of the total trials) that contained joints relevant to the ROM calculation that had frames with a visibility index of 0.5 or below. **Fig 1** (middle) shows a frame from the video capture with the detected joints overlaid on top. The webcam-based solution only collected movement

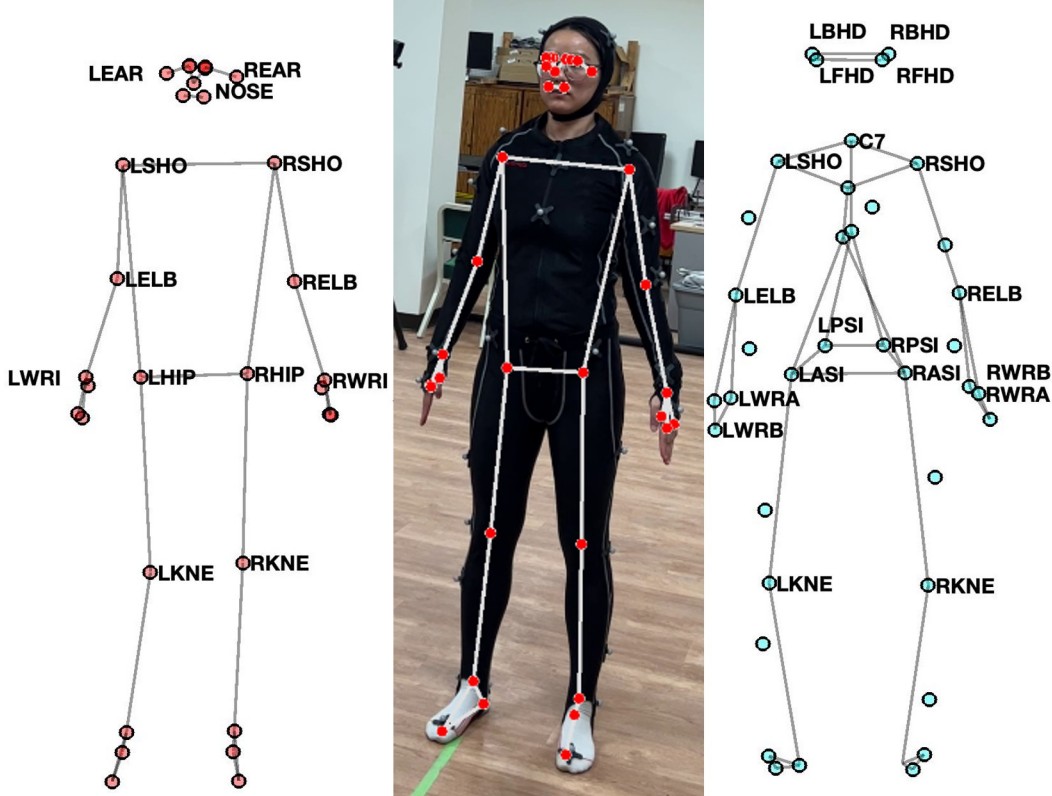

**Fig 1. Landmark positions.** Two-dimensional landmarks for BlazePose (left) and the marker set used for Opti- Track's optical motion capture system (right). BlazePose produces 33 landmarks, whereas OptiTrack's marker set contains 39 markers. Markers relevant to the range of motion calculation are marked with their respective names. Figure in the middle is a screenshot of the landmarks detected by MediaPipe's BlazePose overlayed on a video image captured during data collection with the participant wearing a motion capture suit.

trajectory data from BlazePose while the video feed was not recorded. For the optical motion capture system, OptiTrack Motion Capture System (NaturalPoint, Corvallis, OR, USA) with ten Prime$^x$ 22 cameras was used to record motion at 120 FPS. Data from OptiTrack were streamed to the Python interface through the Motive 2.0 Optical Motion Capture Software and OptiTrack's NatNet SDK. The conventional full-body biomechanical model with 39 markers (**Fig 1**, right) was used to capture joint trajectories (see Figure 2 in [34] for the placement of the entire marker set). The collected data did not contain any identifiable personal information of the participants.

## Procedures

After providing their informed consent, participants were instructed to put on a motion capture suit provided by OptiTrack. Experimenters then fitted the participants with 39 reflective markers based on the locations defined by the full-body model from OptiTrack's Motive software. Subsequently, an experimenter demonstrated each ROM evaluation movement using pre-recorded videos on a tablet computer and the participants were told to repeat the movement as practice. The ROM movements were based on a Range of Joint Motion Evaluation Chart DSHS 13-585A (REV. 03/2014) [35]. Table 1 shows the list of movements used to evaluate the ROM of their corresponding joints, including movements of the spine (extension and flexion, lateral flexion, and trunk rotation), the neck (extension and flexion, lateral bending,

**Table 1. Tested joint range of motion movements.**

| Movement | Recording Orientation | BlazePose Joint 1 | BlazePose Joint 2 | OptiTrack Joint 1 | OptiTrack Joint 2 |
|---|---|---|---|---|---|
| Back Flexion and Extension | Lateral Sagittal | LHIP | LSHO | LPSI | C7 |
| Back Lateral Flexion | Anterior Coronal | LHIP | LSHO | LPSI | C7 |
| Truck Rotation | Anterior Coronal | LSHO | RSHO | LSHO | RSHO |
| Neck Flexion and Extension | Lateral Sagittal | LSHO, RSHO | NOSE | LSHO, RSHO | LFHD, LBHD |
| Neck Lateral Bending | Lateral Sagittal | LSHO, RSHO | NOSE | LSHO, RSHO | LFHD, LBHD |
| Neck Rotation | Anterior Coronal | LEAR | REAR | LFHD, LBHD | RFHD, RBHD |
| Shoulder Adduction and Abduction | Anterior Coronal | LSHO/RSHO | LELB/RELB | LSHO/RSHO | LELB/RELB |
| Shoulder Flexion and Extension | Anterior Coronal | LSHO/RSHO | LELB/RELB | LSHO/RSHO | LELB/RELB |
| Elbow Flexion | Lateral Sagittal | LELB/RELB | LWRI/RWRI | LELB/RELB | LWRA/RWRA, LWRB/RWRB |
| Hip Flexion and Extension | Lateral Sagittal | LHIP/RHIP | LKNE/RKNE | LASI/RASI | LKNE/RKNE |
| Hip Adduction and Abduction | Anterior Coronal | LHIP/RHIP | LKNE/RKNE | LASI/RASI | LKNE/RKNE |

A list of movements used to evaluate the range of motion (ROM) of various joints, the participants' orientation to the webcam during recording, and each movement's corresponding markers for BlazePose and OptiTrack marker sets, and. See S1 Appendix for a breakdown of the acronyms.

and rotation), the upper extremities (shoulder adduction and abduction, shoulder extension and flexion, and elbow extension and flexion), and the lower extremities (hip extension and flexion (knee extended), hip flexion (knee flexed), and hip adduction and abduction). Movements of distal joins, such as wrist and ankle, were excluded from the current study because BlazePose could not reliably generate marker positions of the hands and feet, respectively.

After being familiarized with all movements, participants were guided to stand in front of a computer monitor with a webcam. Participants' standing position coincided with the center of the capture volume of the OptiTrack camera system. All movements were performed while standing upright. To minimize occlusion for the webcam, participants were asked to change their orientation to the camera based on the movements performed. For instance, for shoulder extension and flexion, participants presented the lateral view of their body to the webcam. Additionally, for movements of the lower extremities, participants were holding onto a stool while performing the movements to ensure stability. Each movement was repeated and recorded three times consecutively. For each recording, participants were instructed to perform the movement slowly and deliberately. When the experimenter announced "Start!", the recording would begin, and participants would start the movement. The experimenter would announce "Stop!" and then stop the recording once the participants completed the movement and returned to the upright position.

## Range of motion calculation

The 3D joint trajectory data from OptiTrack and BlazePose were used to derive the joint's ROM angles. In practice, the calculation and extraction of the ROM angle should be implemented in real time and the angular measure should be available upon the completion of a movement. However, due to the experimental nature of the current study, the analysis was performed post-hoc to examine any potential issues that may arise during the data analysis process. Movement trajectory preprocessing was accomplished using TAT-HUM [36], a Python-based movement trajectory analysis toolkit. For trajectory data obtained from BlazePose, points with a low visibility index ($< 0.5$) were removed. Because deriving ROM angle only requires identifying a singular point along the movement trajectory and BlazePose had a low

sampling frequency (15 Hz), data smoothing was deemed unnecessary and was not performed to avoid introducing errors in the filtering process [37, 38].

To derive the ROM angle from a specific movement, two joints are needed to specify the relevant segment. Table 1 shows the corresponding joints for each movement. For some movements, each joint could be specified via a single marker. For instance, back flexion and extension only require the markers placed at the participants' hip (LHIP) and shoulder (LSHO) for BlazePose or at their posterior sacroiliac (LPSI) and C7 for OptiTrack. For other movements, the relevant points may need to be derived as the average location between two markers. For instance, for neck flexion and extension, the nose (NOSE) position was used as one end of the segment whereas the midpoint between the left and right shoulders (LSHO and RSHO) was derived and used as the other end of the segment.

Given the relevant segment, the ROM angle was derived using vector algebra. Let the two points representing the segment be $P_1$ and $P_2$, the normalized vector, $\vec{v}$, for this segment is:

$$\vec{v} = \frac{P_1 - P_2}{\|P_1 - P_2\|}$$

The normalized vectors were calculated for each frame. Then, the movement angle at time $t$, $\alpha_t$, is derived as the angle between the vector at $t$, $\vec{v}_t$ and that at the starting position, $\vec{v}_0$, using dot product:

$$\alpha_t = \arccos\left(\vec{v}_t \cdot \vec{v}_0\right)$$

Subsequently, the final ROM angle can be extracted from the resulting time series of movement angle, defined as the maximum angle during a movement. Although algorithms could easily identify the local maxima of a time series, extraneous factors may introduce noise to the data, rendering this process tricky. For instance, in the resulting time series of movement angle, anomalies may appear due to tracking inconsistency (**Fig 2**). In practice, these anomalies could easily be identified and accounted for via visual inspection in real-time, without affecting the final ROM angle. For the current study, this process was automated using additive seasonality decomposition, which decomposes a signal into seasonal, trend, and residual components, using Python's statsmodels package [39]. In the present context, the trend component corresponds to the change in movement angle. The anomalies could be subsequently identified using the residual values (3 standard deviations from the mean). After removing the anomalies, the ROM angle was identified as the local maximum of all angles. If more than one maximum was identified, visual inspections were performed to identify the appropriate ROM angle.

## Statistical analysis

The three measurements from BlazePose and OptiTrack were used to evaluate their respective test-retest (intra-rater) reliability. Intra-class correlation coefficient (ICC) was computed for each movement with a two-way mixed-effect model for multiple measurements [40, 41] using the ICC package in R [42]. The ICC values can be interpreted as: 0 - 0.2 (slight), 0.2 - 0.4 (fair), 0.4 - 0.6 (moderate), 0.6 - 0.8 (substantial), and 0.8 - 1.0 (almost perfect) [43]. To further illustrate each measure's reliability, the standard error of measurement ($SE_M$) and minimal detectable change (MDC) were also computed. Different from ICC, $SE_M$ represents the measurement error in the same unit as the original measurement [44] and estimates the amount of deviation of repeated measures using the same measurement device from the

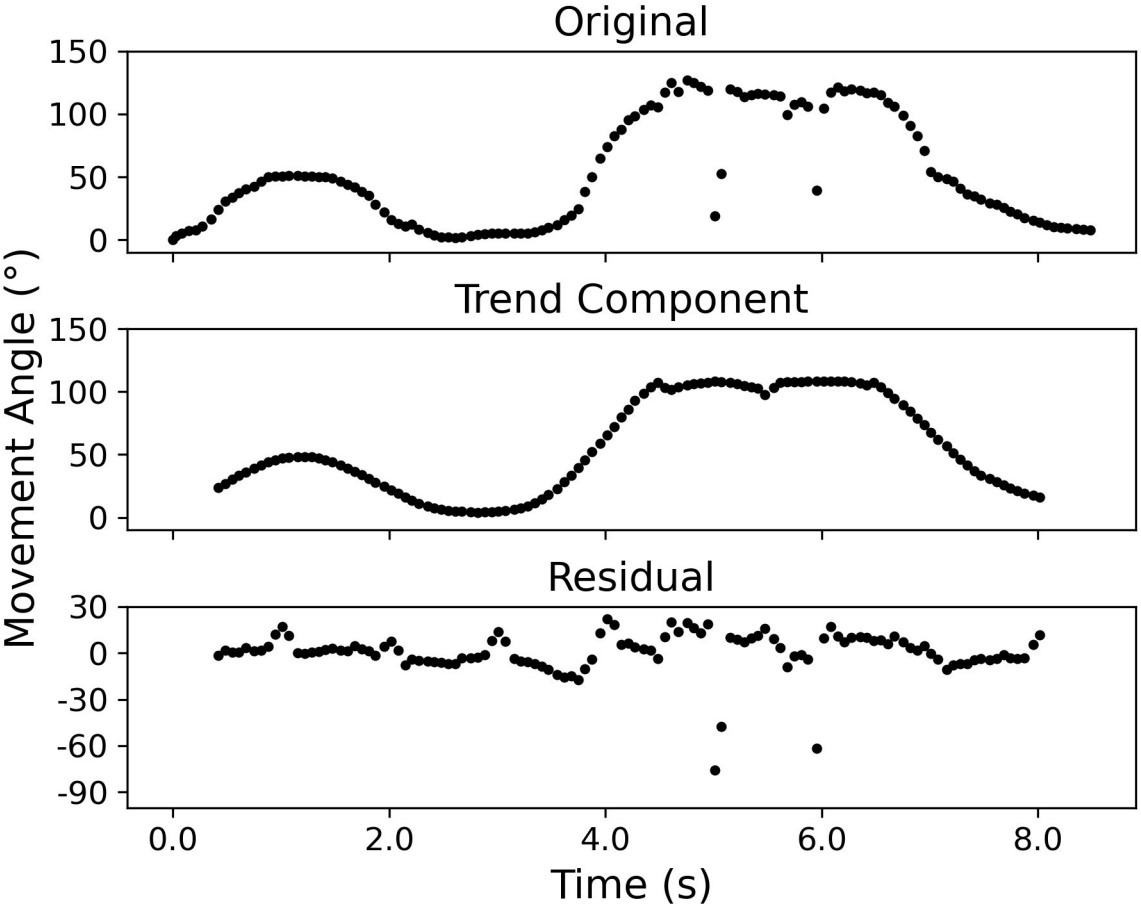

**Fig 2. Anomaly identification.** An illustration of the seasonality decomposition to identify anomalies in the movement angle data. Top panel: The original data contain three noticeable outliers that may affect the subsequent local maxima detection. Middle panel: The derived trend from the seasonality decomposition. Note that the decomposition could potentially introduce artifacts that alter the values of the angle, which was why it was not directly used to identify the maximum angle. Bottom panel: The residuals from the seasonality decomposition where the locations of the three outliers are apparent.

"ground truth", which can be derived using its corresponding ICC:

$$SE_M = \sqrt{\sigma_T^2 \times (1 - ICC)} \qquad \text{Eq (1)}$$

where $\sigma_T^2$ represents the total variance. *MDC* represents the smallest change in value that can be detected beyond random error. *MDC* is based on $SE_M$ and is expressed as

$$MDC = z_{95\%} \times \sqrt{2} SE_M \qquad \text{Eq (2)}$$

where $z_{95\%}$ represents the z-score corresponding to the 95% confidence interval.

Additionally, using results from OptiTrack as the ground truth, the inter-rater reliability between the two measures was also evaluated using two-way mixed-effect ICC for each movement. The individual measurements from BlazePose and OptiTrack were used for the ICC computation, along with the resulting $SE_M$ and MDC. To examine the correlation between measurements from BlazePose and OptiTrack, a linear regression was performed for each joint across all participants, where the OptiTrack measurement was used as the predictor variable and BlazePose measurement was used as the response variable.

## Results

Table 2 shows the test-retest reliability using intraclass correlation coefficient (ICC), standard error of measurement ($SE_M$), and minimal detectable change (MDC) for BlazePose and Opti-Track, as well as their inter-rater reliability using the two-way mixed effect ICC, $SE_M$, and MDC. Overall, results indicate high test-retest and inter-rater reliability for both BlazePose and OptiTrack across different movements.

### Intra-rater reliability

For spine movements (back flexion, back extension, back lateral flexion, and trunk rotation), the 95% confidence intervals (CI) for ICC indicate that the test-retest reliability was almost perfect for BlazePose (low bound from 0.82 to 0.95) with the corresponding $SE_M$ between 1.34˚ and 6.25˚ and the MDC was between 3.71˚ and 17.30˚. Noticeably, even though trunk rotation had a relatively high ICC value (mean = 0.94), its corresponding $SE_M$ and MDC values are relatively high compared to other joints (6.25˚ and 17.30˚, respectively). Because the calculations of $SE_M$ and MDC rely on ICC and the total variability, $\sigma_T^2$ (Eq 1), the discrepancy between $SE_M$/MDC and ICC should be attributed to the high between-subject variability in the measurement. OptiTrack produced similar results for back lateral flexion and trunk rotation where the ICC values were high whereas $SE_M$ and MDC values were low. However, for back flexion and extension, the ICC values were quite low (mean = 0.76 and 0.17, respectively) and $SE_M$ (10.53˚ and 14.79˚) and MDC (29.18˚ and 41.00˚) values were high. This could be attributed to marker instability during the movement. For OptiTrack, back flexion and extension were calculated using markers placed at participants' posterior sacroiliac and C7. When performing the back flexion movement, the participants needed to bend forward with their arms pointing upward. This created tension in the tucked MoCap suit, resulting in marker movement at the posterior sacroiliac. When performing the back extension movement, the participants needed to lean backward. Given the standing position, this would result in partial marker occlusion that had led to inconsistent ROM angle estimation.

For neck movements (extension, flexion, lateral bending, and rotation), the ICC's 95% CI indicates that BlazePose had almost perfect reliability for all movements, where the low bound of the 95% CI ranged from 0.81 to 0.94 and the $SE_M$ and MDC was at a single-digit level. For OptiTrack, all movements had had substantial or almost perfect reliability. The low bound of the ICC's 95% CI ranged between 0.74 (neck flexion) and 0.91 (neck lateral bending). However, compared to BlazePose, OptiTrack's measurement consistency remains relatively low, especially in terms of MDC, which even doubled the amount of BlazePose for neck flexion (12.90˚ for OptiTrack and 6.43˚ for BlazePose).

For the upper extremity (shoulder adduction and abduction, shoulder extension and flexion, and elbow flexion), the test-retest reliability for the ROM measurements from BlazePose was generally high, ranging from substantial to almost perfect reliability. The $SE_M$ and MDC for these movements were generally at around 3˚ and 8˚, respectively. Reliability measures derived for the OptiTrack measurements were comparable to that from BlazePose. Based on the lower bound of the ICC's 95% confidence interval for BlazePose, one joint stood out–elbow flexion (lower bound = 0.67). During data collection, participants had to wear an all-black MoCap suit (**Fig 1** middle). Using a single RGB video stream, BlazePose largely relies on the contrast in the image to perform pose estimation. During elbow flexion, there could be a significant amount of overlap between the forearm and upper arm. Combined with the suit that provides minimum contrast, the overlap presents a non-negligible challenge to the Blaze-Pose algorithm in terms of segmenting and tracking different body parts. Therefore, it is possible that wearing common, everyday clothes (light-colored, form-fitting) that maximize

**Table 2. Intra- and inter-rater reliability measures.**

| Movement | BlazePose | | | OptiTrack | | | BlazePose and OptiTrack | | |
|---|---|---|---|---|---|---|---|---|---|
| | ICC (95% CI) | $SE_M$ (°) | MDC (°) | ICC (95% CI) | $SE_M$ (°) | MDC (°) | ICC (95% CI) | $SE_M$ (°) | MDC (°) |
| **Spine** | | | | | | | | | |
| Back Flexion | 0.98 (0.95, 0.99) | 2.27 | 6.28 | 0.76 (0.53, 0.89) | 10.53 | 29.18 | 0.82 (0.72, 0.89) | 8.11 | 22.49 |
| Back Extension | 0.92 (0.82, 0.96) | 2.10 | 5.83 | 0.17 (-0.65, 0.62) | 14.79 | 41.00 | 0.80 (0.67, 0.88) | 8.07 | 22.36 |
| Back Lateral Flexion | 0.93 (0.88, 0.96) | 1.34 | 3.71 | 0.94 (0.91, 0.97) | 1.50 | 4.15 | 0.92 (0.89, 0.94) | 2.00 | 5.53 |
| Trunk Rotation | 0.94 (0.91, 0.97) | 6.25 | 17.30 | 0.97 (0.95, 0.98) | 3.56 | 9.87 | 0.92 (0.89, 0.94) | 6.96 | 19.31 |
| **Neck** | | | | | | | | | |
| Neck Flexion | 0.93 (0.87, 0.97) | 2.32 | 6.43 | 0.87 (0.74, 0.94) | 4.65 | 12.90 | 0.92 (0.88, 0.95) | 3.45 | 9.56 |
| Neck Extension | 0.90 (0.81, 0.96) | 3.24 | 8.99 | 0.91 (0.82, 0.96) | 3.73 | 10.33 | 0.89 (0.83, 0.93) | 4.22 | 11.69 |
| Neck Lateral Bending | 0.96 (0.94, 0.98) | 1.44 | 4.00 | 0.94 (0.91, 0.97) | 2.22 | 6.16 | 0.81 (0.73, 0.86) | 4.77 | 13.21 |
| Neck Rotation | 0.95 (0.91, 0.97) | 3.46 | 9.58 | 0.85 (0.77, 0.91) | 3.99 | 11.05 | 0.62 (0.47, 0.73) | 11.57 | 32.08 |
| **Upper Extremity** | | | | | | | | | |
| Shoulder Adduction | 0.92 (0.87, 0.95) | 3.04 | 8.41 | 0.95 (0.92, 0.97) | 2.53 | 7.02 | 0.89 (0.84, 0.92) | 3.98 | 11.03 |
| Shoulder Abduction | 0.97 (0.95, 0.98) | 2.41 | 6.68 | 0.81 (0.7, 0.89) | 7.98 | 22.11 | 0.68 (0.56, 0.77) | 12.43 | 34.44 |
| Shoulder Flexion | 0.94 (0.90, 0.96) | 1.66 | 4.61 | 0.82 (0.72, 0.9) | 6.32 | 17.52 | 0.18 (-0.14, 0.40) | 9.73 | 26.97 |
| Shoulder Extension | 0.93 (0.89, 0.96) | 3.17 | 8.78 | 0.93 (0.89, 0.96) | 2.60 | 7.20 | 0.92 (0.88, 0.94) | 3.27 | 9.07 |
| Elbow Flexion | 0.79 (0.67, 0.87) | 3.76 | 10.4 | 0.83 (0.73, 0.9) | 6.68 | 18.5 | 0.53 (0.35, 0.66) | 11.69 | 32.40 |
| **Lower Extremity** | | | | | | | | | |
| Hip Flexion | 0.94 (0.91, 0.97) | 2.60 | 7.22 | 0.94 (0.90, 0.96) | 2.65 | 7.34 | 0.83 (0.76, 0.88) | 5.18 | 14.35 |
| Hip Extension | 0.95 (0.91, 0.97) | 2.87 | 7.95 | 0.94 (0.90, 0.96) | 2.18 | 6.03 | 0.84 (0.78, 0.88) | 4.27 | 11.83 |
| Hip Flexion (Knee Flexed) | 0.85 (0.76, 0.91) | 3.06 | 8.49 | 0.86 (0.77, 0.92) | 3.68 | 10.21 | 0.64 (0.50, 0.74) | 5.67 | 15.71 |
| Hip Adduction | 0.93 (0.88, 0.96) | 2.65 | 7.35 | 0.88 (0.8, 0.93) | 3.22 | 8.94 | 0.89 (0.84, 0.92) | 3.38 | 9.37 |
| Hip Abduction | 0.93 (0.89, 0.96) | 3.22 | 8.94 | 0.95 (0.92, 0.97) | 2.95 | 8.19 | 0.95 (0.94, 0.97) | 2.68 | 7.43 |

The intraclass correlation coefficient (ICC), standard error of measurement ($SE_M$), and minimal detectable change (MDC) for the range of motion angles derived using BlazePose and OptiTrack. ICC interpretations: 0 - 0.2 (slight), 0.2 - 0.4 (fair), 0.4 - 0.6 (moderate), 0.6 - 0.8 (substantial), and 0.8 - 1.0 (almost perfect).

contrast under natural lighting during the collection process could significantly improve the reliability of these movements. Nonetheless, comparing between BlazePose and OptiTrack, the latter again shows lower consistency with relatively high $SE_M$ and MDC values.

Finally, for the lower extremity (hip flexion and extension (knee extended), hip flexion (knee flexed), and hip adduction and abduction), the ICC values for BlazePose were generally high except for hip flexion (knee flexed) (95% CI low bound = 0.76). This could again be attributed to the issue of occlusion combined with a lack of contrast in the clothing as in the case of elbow flexion. All the other measures were comparable to those of other joints. Measurements derived using OptiTrack data produced comparable reliability measures as those from BlazePose.

In summary, the intra-rater reliability of BlazePose's ROM calculation is relatively high and comparable to that of OptiTrack. Although the intra-class ICC values for some joints are relatively low, it could be an artifact of the experimental setup, such as the low contrast MoCap suits. These issues could be easily addressed in practice to further improve the reliability of the ROM measurement. More critically, the analysis also revealed that although OptiTrack could provide accurate and precise marker positions, the markers' placement could be perturbed during the movement as the markers were attached to an elastic MoCap suit. Although alternative marker placement is possible to minimize the perturbation to the markers' placement such as directly attaching the markers to the participant's skin), these methods could be invasive and impractical for ROM evaluation.

### Inter-rater reliability

Despite their respective high intra-class ICC, the mixed-effect ICC between BlazePose and OptiTrack varies notably from joint to joint. On the one hand, some movements have relatively higher ICC values, such as back lateral flexion (ICC = 0.92, CI = [0.89, 0.94]), trunk rotation (ICC = 0.92, CI = [0.89, 0.94]), neck flexion (ICC = 0.92, CI = [0.88, 0.95]), shoulder adduction (ICC = 0.89, CI = [0.84, 0.92]), shoulder extension (ICC = 0.92, CI = [0.88, 0.94]), and hip abduction (ICC = 0.89, CI = [0.84, 0.92]) and adduction (ICC = 0.95, CI = [0.94, 0.97]). On the other hand, some movements have extremely low ICC values, such as shoulder flexion (ICC = 0.18, CI = [-0.14, 0.40]), elbow flexion (ICC = 0.53, CI = [0.35, 0.66]), and hip flexion (knee flexed) (ICC = 0.64, CI = [0.50, 0.74]).

**Fig 3** presents the regression between the ROM angles from OptiTrack and BlazePose for each joint. The regression goodness of fit reflects the ICC results, where joints with high intra-class ICC also have relatively high $r^2$, such as black flexion ($r^2 = 0.77$), trunk rotation ($r^2 = 0.75$), neck flexion ($r^2 = 0.79$), shoulder abduction ($r^2 = 0.75$), and hip abduction ($r^2 = 0.84$), and vice versa for those with low intra-class ICC, such as shoulder flexion ($r^2 = 0.03$), elbow flexion ($r^2 = 0.29$), and hip flexion (knee flexed) ($r^2 = 0.23$). The regression results revealed the reason behind the low inter-class correlation for some joints. Specifically, for shoulder flexion, elbow flexion, and hip flexion (knee flexed), the relationship between OptiTrack and BlazePose remains relatively flat, where BlazePose's estimated ROM angle remained unchanged as the OptiTrack estimate varied for different participants. **Fig 4** compares two movement angle samples of elbow flexion, where the derived ROM angle difference between the OptiTrack's two samples was 33˚ whereas that between the BlazePose's samples was only 14˚. Noticeably, the figure shows that BlazePose's movement angle remained relatively flat at the apex of the trajectory, whereas the OptiTrack's trajectory is more parabolic. This observation suggests that BlazePose may not be as sensitive to the joint locations as the elbow reaches its maximum flexion position.

## Discussion

The current study presented a webcam-based machine learning solution using BlazePose for range of motion (ROM) measurement and evaluated its intra- and inter-rater reliability as compared to a marker-based motion capture system, OptiTrack. Unlike previous studies, this study examined a diverse set of joints using a full-body biomechanical model, including those of the spine, the neck, and the upper and lower extremities.

Results revealed high intra-rater reliability for BlazePose and OptiTrack. For almost all movements, both measurement methods had substantial or almost perfect intra-rater reliability, with their corresponding measurement errors ($SE_M$) below 5˚ and the minimal detectable change (MDC) at around 10˚. This indicates that the webcam-based solution could reliably differentiate changes in ROM from measurement errors when the amount of change is greater than 10˚. From a practitioner's perspective, this could be sufficient to evaluate the effect of an intervention, such as in the case of cervical joint [45]. Future studies could focus on systematically delineating BlazePose's sensitivity to changes in ROM angles for each joint. Furthermore, the examination of movements with lower inter-rater reliability suggests that changing the participants' orientation to the camera's line of sight and the color scheme of their cloth could further improve BlazePose's inter-rater reliability. Future studies could focus on developing the best practice when collecting ROM data using BlazePose.

Despite their respective high test-retest reliability, the inter-rater reliability between Blaze-Pose and OptiTrack is variable across different joints, with some joint movements having relatively low inter-rater reliability. Upon close examination of these movements, it became evident that the variability in inter-rater reliability could be attributed to BlazePose's reduced

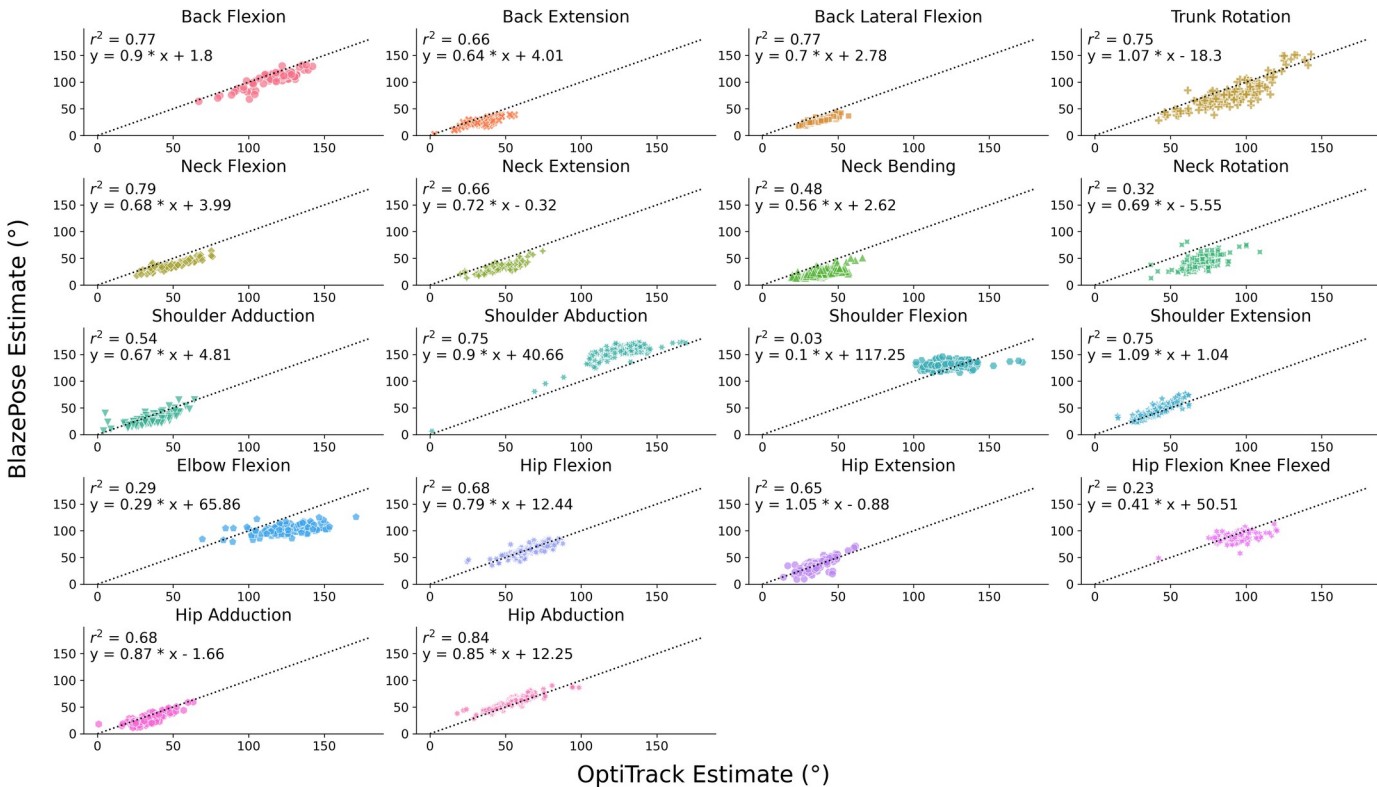

**Fig 3. Joint-based regression.** Regressions between the range of motion (ROM) angles derived from OptiTrack (x-axis) and BlazePose (y-axis) across all participants for each joint. Different markers represent different joint movements. The dotted lines in each subplot are reference lines and have a slope of 1 and an intercept of 0.

sensitivity to joint locations as the movement angle reaches its apex (**Fig 4**). This diminished sensitivity is likely due to the lack of contrast in the participants' clothing and other environmental factors, which hampers the algorithm's ability to accurately identify joint locations. **Fig 5** shows an experimenter performing the elbow flexion movement captured at the starting and maximally flexed positions. At the starting position, BlazePose effectively identifies the appropriate locations of the elbow and wrist joints. However, as the elbow joint reaches maximal flexion, the estimated locations of these joints deviate from their actual positions. Notably, BlazePose's forearm angle appears noticeably smaller than the actual angle in the figure. Such inaccuracies in joint location identification could result from various factors, including the lack of contrast in the MoCap suit and the crowded testing environment. To address these issues, future studies should systematically investigate the effectiveness of BlazePose in recovering joint locations and establish detailed guidelines for an optimal setup to enhance ROM evaluation. By doing so, the reliability and accuracy of BlazePose as a tool for joint movement assessment can be improved.

From a practitioner's perspective, the effectiveness of a ROM measurement method should be evaluated from a functional standpoint. In this context, the measurement method should not solely focus on achieving high anatomical accuracy and precision. Instead, the method's intra-rater reliability and accessibility are also equally important. Specifically, ROM is commonly used to evaluate and quantify the improvement in a patient's mobility at a certain joint because of an intervention. Therefore, an ideal ROM measurement method should reliably evaluate the ROM across multiple sessions. Additionally, accessibility is also a critical

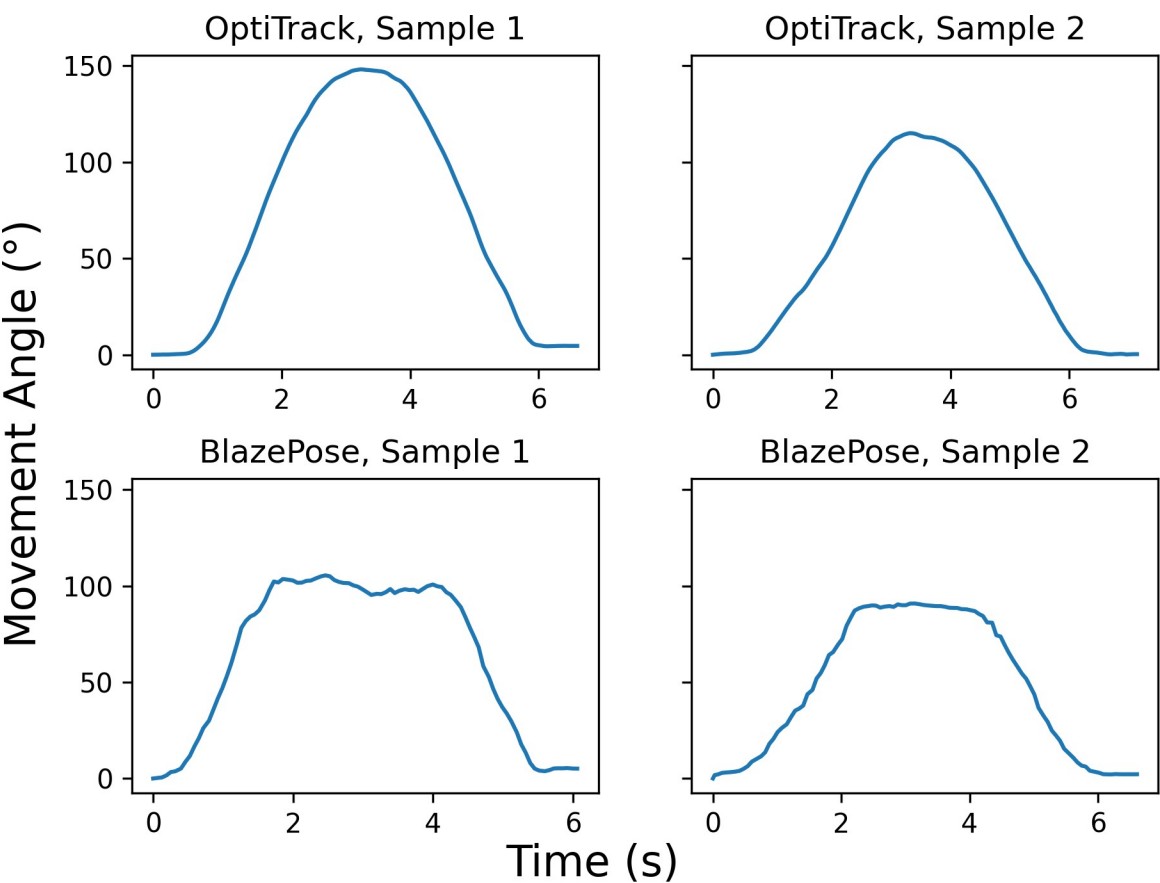

**Fig 4. Sample movement angles of elbow flexion.** Sample movement angles as a function of time for elbow flexion from OptiTrack (top row) and BlazePose (bottom row).

consideration. The COVID-19 pandemic raises awareness of the importance of telehealth and its effectiveness in lowering the barrier to access for disadvantaged populations. In the context of physical therapy, ROM evaluation has always relied on a goniometer [4, 5], which has long been known for its lack of reliability despite requiring extensive training and practice [11, 12, 46]. Merely relying on a goniometer for physical therapy would disproportionately affect patients of different racial and socioeconomic backgrounds.

As an alternative, the webcam-based ROM evaluation tool presented in the current study could potentially be more accessible compared to a goniometer. Since BlazePose is lightweight and compatible with multiple platforms, it is possible to adapt the current data collection and processing pipeline to a browser-based web application and mobile applications on iOS and Android devices. With this implementation, users would simply need to appropriately orient themselves in front a webcam on a home computer to have their ROM measured, which would reduce the needs for in-person ROM evaluation using a goniometer. Importantly, to produce accurate and reliable measurements, this alternative solution may be more effective in estimating ROM angles for certain joints than others and it also requires appropriate camera placement. To the best of our knowledge, this study is the first one that applies an open-access pose-estimation algorithm to measure the ROM of a diverse set of joints. Future studies could capitalize on the results of this study and focus on a subset of joints to investigate the measurement's reliability and accuracy, as well as the ideal setup for a home environment.

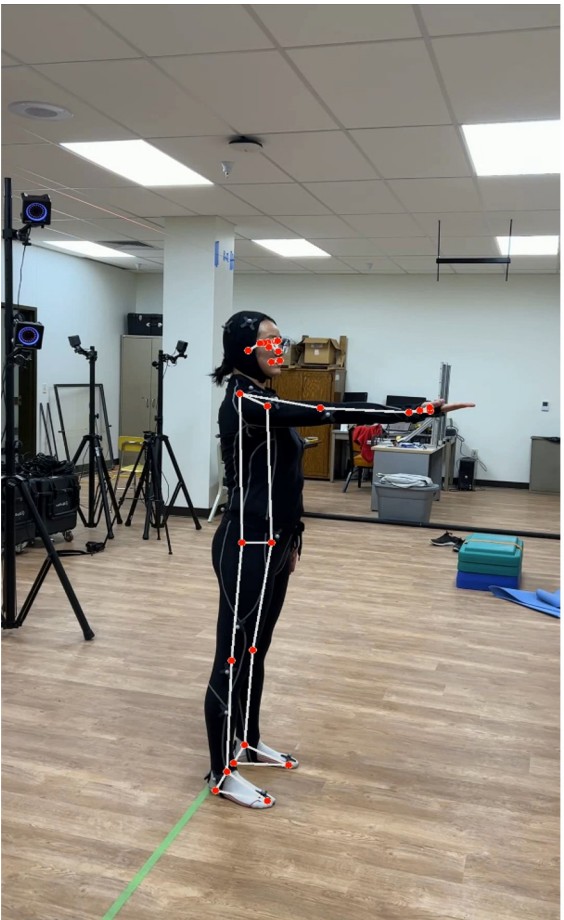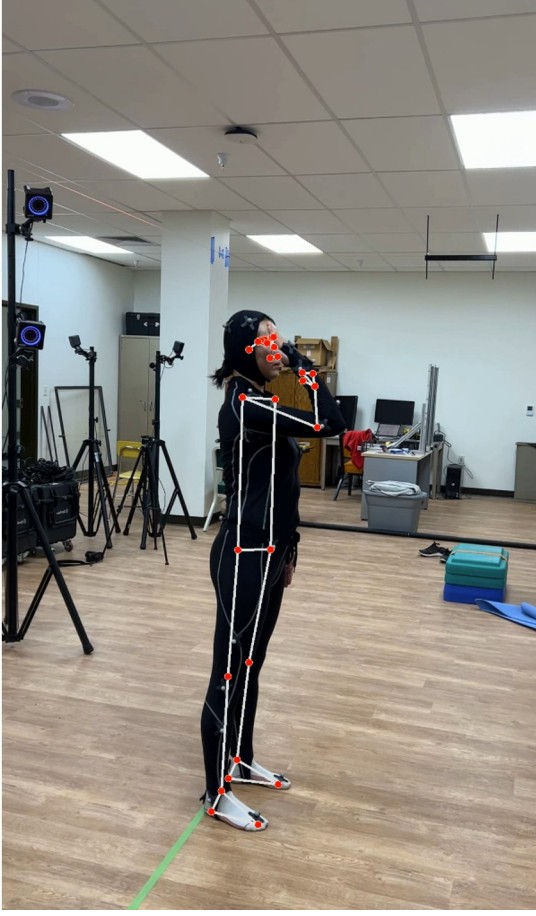

**Fig 5. Elbow flexion demo.** An experimenter performing the elbow flexion movement at the starting (left) and maximally flexed (right) positions with an overlay of BlazePose's estimated joint locations.

## Limitations

As a proof of concept, the present study demonstrated the potential of adopting a digital solution for ROM evaluation via a single webcam. However, there are a few limitations to the current study that future works could improve upon.

First and foremost, while the goniometer is a widely used method for ROM evaluation, it was not included in the reliability comparison of the current study. This decision was prompted by practical challenges, as the researchers did not have access to certified and experienced clinicians who could accurately and reliably perform goniometer-based ROM evaluations. The absence of a reliable researcher may have compromised the accuracy of the resulting measurements, which prevented the current study from incorporating the goniometer measure. For future studies, it is essential to collaborate with professional practitioners to conduct a comprehensive comparison of reliability analysis between the BlazePose and goniometer measurements. This approach will yield more robust and practical results for ROM evaluations.

Second, given the exploratory nature of the present study, the ROM evaluation tool developed for this research demands technical proficiency, particularly in Python. As mentioned earlier, the MediaPipe framework and BlazePose offer a range of application programming

interfaces (APIs), including JavaScript for web browsers, and iOS and Android for mobile applications. Leveraging these APIs presents an opportunity to create standalone ROM evaluation applications with intuitive graphical user interfaces (GUIs). Such user-friendly applications would not necessitate technical expertise, making them more accessible and usable for a wider audience.

Thirdly, the current study extensively covered a broad range of joint movements; however, movements involving distal joints, such as ankle inversion and eversion, ankle flexion and extension, wrist radial and ulnar, wrist extension and flexion, as well as forearm pronation and supination (as included in [35]), were excluded. This decision was attributed to BlazePose's lack of consistent hand and foot tracking. In the given setup, the hands and feet occupy a relatively smaller portion of the single image (as shown in **Fig 5**) making it more challenging to accurately track pose landmarks and leading to inconsistent angle measurements. While the MediaPipe framework offers alternative solutions for robust and accurate hand pose estimation (as demonstrated in [47]), the same level of accuracy for foot tracking is still lacking. For future studies, incorporating hand pose estimation could enable ROM evaluation of hand joints and help explore whether closer views of the feet would improve tracking consistency. By addressing these limitations, researchers can achieve a more comprehensive assessment of joint movements and enhance the applicability of BlazePose in diverse scenarios.

Finally, it's important to note that the current study exclusively recruited healthy adults who could perform various ROM movements in a standing posture. However, in practical applications, this tool will be used by patients who may have physical frailty or neurodivergent conditions, making it challenging for them to achieve the standing position used in this study. To address this issue, future collaborations with professional practitioners are essential. Such collaborations would allow the development of a protocol that provides recommendations on the optimal recording position for the clinical population in which ROM evaluation is to be performed. By considering the needs of these individuals, the tool's usability and applicability can be extended to a broader range of patients.

## Conclusion

In conclusion, the current study presents an alternative way of measuring joint range of motion (ROM) that merely relies on a webcam setup. Compared to an optical motion capture system, the webcam-based machine learning approach demonstrated high intra- and inter-rater, as well as individual-level reliability in quantifying and assessing the ROM of some joints. Further developing this tool to improve its reliability for other joints and eventually adopting it for tele-implementation of physical therapy and rehabilitation could significantly reduce the barrier to access to healthcare.

## Supporting information

**S1 Appendix. Joint acronym lookup.**
(DOCX)

## Acknowledgments

We would like to thank Dr. Justin Huber and another reviewer for their thoughtful comments during the review process.

## Author Contributions

**Conceptualization:** Xiaoye Michael Wang, Derek T. Smith, Qin Zhu.

**Data curation:** Xiaoye Michael Wang.

**Formal analysis:** Xiaoye Michael Wang.

**Funding acquisition:** Derek T. Smith, Qin Zhu.

**Investigation:** Xiaoye Michael Wang, Qin Zhu.

**Methodology:** Xiaoye Michael Wang, Derek T. Smith, Qin Zhu.

**Project administration:** Xiaoye Michael Wang, Qin Zhu.

**Resources:** Xiaoye Michael Wang, Derek T. Smith, Qin Zhu.

**Software:** Xiaoye Michael Wang.

**Supervision:** Derek T. Smith.

**Validation:** Xiaoye Michael Wang.

**Visualization:** Xiaoye Michael Wang.

**Writing – original draft:** Xiaoye Michael Wang.

**Writing – review & editing:** Xiaoye Michael Wang, Derek T. Smith, Qin Zhu.

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
