## [Decision Letter · Decision Letter 0]

4 Jul 2023

PONE-D-23-15143A webcam-based machine learning approach for three-dimensional range of motion evaluationPLOS ONE

Dear Dr. Wang,

Thank you for submitting your manuscript to PLOS ONE. After careful consideration, we feel that it has merit but does not fully meet PLOS ONE’s publication criteria as it currently stands. Therefore, we invite you to submit a revised version of the manuscript that addresses the points raised during the review process.

Thank you for submitting your manuscript to PLOS ONE. The reviewers have made a number of helpful suggestions to consider as you revise your manuscript. Please be sure to address all reviewer comments in a point-by-point response document. We look forward to receiving your revised manuscript!

We look forward to receiving your revised manuscript.

Kind regards,

Ryan T. Roemmich

Academic Editor

PLOS ONE

Journal Requirements:

"This study was supported by the National Institute on Minority Health and Health Disparities (STTR: #1R41MD015689-01), and the US Department of the Treasure and Wyoming Health & Bioscience Innovation Hub (CARES-HUB-MOVE#1040)."

Additional Editor Comments:

Thank you for submitting your manuscript to PLOS ONE. The reviewers have made a number of helpful suggestions to consider as you revise your manuscript. Please be sure to address all reviewer comments in a point-by-point response document. We look forward to receiving your revised manuscript!

Reviewers' comments:

Reviewer's Responses to Questions

**Comments to the Author**

1. Is the manuscript technically sound, and do the data support the conclusions?

Reviewer #1: Yes

Reviewer #2: Partly

2. Has the statistical analysis been performed appropriately and rigorously? 

Reviewer #1: N/A

Reviewer #2: Yes

3. Have the authors made all data underlying the findings in their manuscript fully available?

Reviewer #1: No

Reviewer #2: Yes

4. Is the manuscript presented in an intelligible fashion and written in standard English?

Reviewer #1: Yes

Reviewer #2: Yes

5. Review Comments to the Author

Reviewer #1: Summary:

In this work, the authors evaluate a webcam-based method for measuring range of motion (ROM) including flexion, extension, and rotation of the neck, back, and upper and lower extremities. The authors compare their proposed method to a motion-capture system (OptiTrack), evaluating the ICC, standard error of measurement (SEM), and minimal detectable change (MDC) within each system, as well as between the systems. The methods are evaluated on a cohort of 25 university students.

General Comments:

The paper is well-written and presents interesting results with respect to the performance of a webcam-based method for ROM assessment. The authors note good to excellent ICC within multiple trials of the webcam method for a given participant. However, the ICC between the webcam and OptiTrack varied significantly by each joint. The authors attribute this to differences in keypoint/marker placement between the methods, and concerns with lack of contrast due to the dark clothing that must be worn during motion capture.

Specific Comments:

1. Please provide more detail in the Abstract with respect to the method, the joints assessed, and specific key findings.

2. Please do not capitalize “Pandemic” throughout the manuscript.

3. Lines 64 – 66: Consider removing CNNs from this sentence and discussing this separately. Currently the other approaches listed in this sentence are different modalities for collecting data, whereas CNNs can be used in conjunction with any of these as a means to process and form meaningful conclusions from the collected data.

4. Lines 69 – 70: Please also include the need for a large physical space to perform motion capture as a limitation of this approach.

5. Lines 70 – 71: Please change “In recent years, the machine learning-based human pose estimation has seen significant development.(19–21)” to “In recent years, the machine learning-based human pose estimation field has seen significant development.(19–21)” or similar.

6. Lines 83 – 89: Please also add gait assessment as a field where the use of human-pose estimation has been explored.

7. The study in this work focused on multiple joints in the upper body: J. Fan et al., “Reliability of a human pose tracking algorithm for measuring upper limb joints: comparison with photography-based goniometry,” BMC Musculoskelet Disord, vol. 23, no. 1, p. 877, 2022, doi: 10.1186/s12891-022-05826-4.

8. Section 2.2: Please provide the name of the motion capture system in this section, as well as the sampling frequency of both the motion capture system and the webcam.

9. Methods - General: Was the webcam placed a prescribed distance from the participant for each movement?

10. Table 1: Please specify the orientation of the camera/plane from which the video was taken for each movement.

11. Lines 172 – 174: Please justify why a different cut-off frequency was used for the webcam and motion capture system.

12. Lines 236 – 238: Were the inter-rater ICC results significantly different when the measurements from BlazePose and OptiTrack were not averaged over the 3 trials? If possible, it may be valuable to investigate whether the process of averaging has affected the results in a significant way.

13. Lines 238 – 240: Please review this sentence, as it does not appear to be complete.

14. Table 2: Please provide units in the table headings where applicable (ie for SEM and MDC).

15. Lines 270 – 274: Was this hypothesis supported by a higher percentage of timesteps where visibility <0.5 for the movements with lower ICC? It may be valuable to provide an summary of the percentage of timesteps where the visibility was less than the 0.5 cut-off (in a supplemental document or the main manuscript).

16. Section 3.3: It would also be valuable to see correlations by-joint (potentially in the supplemental material)

17. Lines 342 – 343: While the mean slope is 1.01, the range is [0.83 – 1.13] which suggests the mapping between the two modalities is not always “one-to-one” as the authors suggest. Presenting an analysis by-joint may provide further insight with respect to whether there are any joints where the mapping between the two systems is better or worse.

18. Figure 4a): Please ensure that the right-most edge of the figure is not cropped out/obstructed.

19. Lines 383 – 384: Reference (41) focuses solely on the cervical joint. It is unclear whether the MDC of 10 degrees applies to all joints studied in this work.

20. Lines 384 – 387: “Careful examination” of the movements with lower inter-rater reliability as claimed in this sentence has not been presented in this manuscript. Further examination to support this hypothesis is required to make this claim. Please reword this sentence if further analysis on this hypothesis is not possible.

21. Line 416: “appropriate camera placement” is a key component of the webcam-based method presented in this work. Please provide additional detail as to the height and distance of the camera to the participant, as well as how the videos were framed for each movement.

22. Line 425: Based on the results in Table 2, it is too strong of a claim to state that there was high inter-rater reliability across all joints. Please rephrase this to acknowledge that inter-rater reliability was low for some joints.

Reviewer #2: Overall Impression:

- This study explored the use of a single-camera approach to obtaining human biomechanics by using a machine-learning solution (Blazepose) based on human pose detection. The reliability of this approach is studied in a sample of 25 adult participants, and the accuracy of the approach is compared to that of a marker-based motion capture system. While I’m excited to see validation studies on such systems, I recommend further revisions to align author’s conclusions with the questions tested in this study.

Major issues:

- Recommend caution with claim of cost-effectiveness (and accessibility) as this does not appear to be a factor tested in this paper. Without explicit testing, the cost/accessibility of a device-based Blazepose approach to ROM measurement would seem more than that of a simple goniometer.

- There is no limitations section in paper.

- Statistical methods seem to be duplicative. For example, the authors use inter-rater reliability, linear regression and root-mean-square-error to describe the accuracy of Blazepose method versus motion capture method.

Minor issues:

- Beginning line 34, the abstract would benefit from a concise statement of actual findings regarding reliability and accuracy rather than qualitative statements alone.

- Line 70, authors state that technical proficiency is a barrier to motion capture. Would there not be some technical barriers to implementing the Blazepose solution? While hopefully less barriers, consider acknowledging (in a limitations section) that some technical hurdles still remain for this method.

- Typo on line 84, “have been used” instead of “has been used”

- Typo on line 87, “examines” instead of examine

- Line 94, while motion capture is an acceptable comparator, the authors have stated the goniometer is one of the most common ways to measure ROM. I would then expect to see this as a comparator in the study.

- Line 95, the last sentence of section claims Blazepose as an alternative to traditional ROM evaluation. Are authors referring to the goniometer as traditional method? How have the authors tested the accessibility of Blazepose versus a goniometer? This sentence would benefit from revision, and contents of sentence would be better suited in results section.

- Line 103, the methods section would benefit from more details on participant demographics. Were all participants healthy / neurotypical? Were any neurodivergent populations included? Were any amputees included?

- Typo on line 127, “were used” instead of “was used”

- Line 130, if webcam video of participants, was face occlusion used to prevent identification? Based on Figure 1, the red dots provide some occlusion but I would be concerned that the remainder of visible face could potentially be recognized and person identified.

- Line 143, please clarify whether the ROM evaluation movements are based on a standard or on some other citable reference. Also, it might benefit the audience to explain/justify why more distal joint movements were excluded, e.g. wrist and ankle ROM.

- Line 152, the standing and repositioning required for the webcam could be quite challenging for some participants, especially individuals who are physically frail or neurodivergent. These populations often require therapy interventions to address function/quality of life. While I’m excited to see these pose detection systems and expect future iterations will be more robust to human movement in-the-wild, the current system described does have limitations which are important for clinicians to understand.

- Line 157, for repeated measures of each movement, were these measures all performed during the same session? If so, was there a prescribed time period between measures?

- Line 173, please provide some explanation or reference regarding the choice of pre-processing filter designs. Would cutoff frequency of 5Hz risk filtering out the frequency of some human movements?

- Beginning at Line 198, the methods to calculate ROM seem overly complex and the justification is confusing. Using motion capture or the Blazepose method, I would expect these to yield joint locations in 3D, which could then be used to define vectors (e.g. for elbow angle, a vector from shoulder-to-elbow would exist and a vector from elbow-to-wrist would exist). The authors could then determine ROM by calculating the angle between vectors at every instance in time (as the authors later describe). While the pivot joint is rotating and translating with time as the authors depict in Figure 2, the ability to determine the angle at each instance in time would make this rotation/translation seemingly a non-factor. Please clarify further (or highlight references) to support using best-fit planes and joints centroids to calculate the ROM.

- Line 327, the authors report that compression around the acromion caused the Optitrak shoulder location to be perturbed. Given that the 3D coordinate data is available, please quantify the perturbation for the reader. Based on Figure 3a, this perturbation visually appears quite small, and its contribution to the higher interrater reliability in the shoulder flexion ROM would also seem small. Likewise, as 3D coordinate data is available, please quantify the “lack of perturbation” of the shoulder marker during shoulder extension movement.

- Line 412, the authors state the webcam approach is affordable and accessible, but these must be interpreted relatively. Do the authors suggest webcam is more affordable/accessible than goniometer? Or motion capture? Furthermore, as affordability/accessibility were not tested in the described study, I suggest these statements be revised.

6. PLOS authors have the option to publish the peer review history of their article (what does this mean?). If published, this will include your full peer review and any attached files.

Reviewer #1: No

Reviewer #2: **Yes: **Justin Huber

---

## [Author Response · Author response to Decision Letter 0]

18 Aug 2023

Reviewer 1

The paper is well-written and presents interesting results with respect to the performance of a webcam-based method for ROM assessment. The authors note good to excellent ICC within multiple trials of the webcam method for a given participant. However, the ICC between the webcam and OptiTrack varied significantly by each joint. The authors attribute this to differences in keypoint/marker placement between the methods, and concerns with lack of contrast due to the dark clothing that must be worn during motion capture.

We thank the reviewer for their positive evaluation of our work. We hope that our revision would help to address the issues that the reviewer found in its original version. 

Specific Comments:

1. Please provide more detail in the Abstract with respect to the method, the joints assessed, and specific key findings.

We added a description of the actual findings. 

2. Please do not capitalize “Pandemic” throughout the manuscript.

We changed all mentions of pandemic to lowercase. 

3. Lines 64 – 66: Consider removing CNNs from this sentence and discussing this separately. Currently the other approaches listed in this sentence are different modalities for collecting data, whereas CNNs can be used in conjunction with any of these as a means to process and form meaningful conclusions from the collected data.

We removed CNN from this sentence and moved it to the second half of the paragraph. 

4. Lines 69 – 70: Please also include the need for a large physical space to perform motion capture as a limitation of this approach.

We added the need for a large physical space as an additional limitation of this approach. 

5. Lines 70 – 71: Please change “In recent years, the machine learning-based human pose estimation has seen significant development.(19–21)” to “In recent years, the machine learning-based human pose estimation field has seen significant development.(19–21)” or similar.

We changed the sentence to “In recent years, the field of machine learning-based human pose estimation has seen significant development. (19–21)”

6. Lines 83 – 89: Please also add gait assessment as a field where the use of human-pose estimation has been explored.

We added gait assessment to the list. 

7. The study in this work focused on multiple joints in the upper body: J. Fan et al., “Reliability of a human pose tracking algorithm for measuring upper limb joints: comparison with photography-based goniometry,” BMC Musculoskelet Disord, vol. 23, no. 1, p. 877, 2022, doi: 10.1186/s12891-022-05826-4.

We thank the reviewer for pointing out this relevant work. We modified the sentence on lines 88-89 to reflect the literature. 

8. Section 2.2: Please provide the name of the motion capture system in this section, as well as the sampling frequency of both the motion capture system and the webcam.

The name of the motion capture system was provided in the original text “OptiTrack Motion Capture System (NaturalPoint, Corvallis, OR, USA) with ten Primex 22 cameras”. Information of each system’s sampling frequency was added to the text. 

9. Methods - General: Was the webcam placed a prescribed distance from the participant for each movement?

The webcam was placed at approximately 4 m away from the participant. This is to ensure that the participant’s entire body was visible to the webcam. This detail was added to the manuscript. 

10. Table 1: Please specify the orientation of the camera/plane from which the video was taken for each movement.

An additional column was added to specify each movement’s recording orientation. 

11. Lines 172 – 174: Please justify why a different cut-off frequency was used for the webcam and motion capture system.

Different cutoff frequencies were used in response to the different sampling frequencies of BlazePose and OptiTrack. This detail was added to the text. 

12. Lines 236 – 238: Were the inter-rater ICC results significantly different when the measurements from BlazePose and OptiTrack were not averaged over the 3 trials? If possible, it may be valuable to investigate whether the process of averaging has affected the results in a significant way.

We thank the reviewer for bringing up the issue of inter-rater ICC. After careful considerations, we decided to use individual measurements, instead of the average measurements, to calculate the inter-rater ICC. 

The motivation behind this could be attributed to Reviewer 2, whose comments focus more on the implication of this ROM evaluation tool in practice while working with actual patients. In this context, expecting the patients to repeat the same ROM movements for three consecutive times may be unrealistic and, therefore, presenting the cross-validation results based on the averaged data may not truthfully reflect what would be needed in practice. 

We edited the manuscript to reflect this change. 

13. Lines 238 – 240: Please review this sentence, as it does not appear to be complete.

Thanks for pointing out this error. We completed the sentence. 

14. Table 2: Please provide units in the table headings where applicable (ie for SEM and MDC).

Units are added to the table headers.

15. Lines 270 – 274: Was this hypothesis supported by a higher percentage of timesteps where visibility <0.5 for the movements with lower ICC? It may be valuable to provide an summary of the percentage of timesteps where the visibility was less than the 0.5 cut-off (in a supplemental document or the main manuscript).

In only considering the relevant joints for a specific ROM calculation, instances in which the low visibility index occurred were really rare (4 trials, or 0.13% of the total trials). This detail was added to the text. 

16. Section 3.3: It would also be valuable to see correlations by-joint (potentially in the supplemental material)

We removed the RMSE figure and substituted it with the by-joint correlations. 

17. Lines 342 – 343: While the mean slope is 1.01, the range is [0.83 – 1.13] which suggests the mapping between the two modalities is not always “one-to-one” as the authors suggest. Presenting an analysis by-joint may provide further insight with respect to whether there are any joints where the mapping between the two systems is better or worse.

We added the by-joint correlations. 

18. Figure 4a): Please ensure that the right-most edge of the figure is not cropped out/obstructed.

The figure is fixed. 

19. Lines 383 – 384: Reference (41) focuses solely on the cervical joint. It is unclear whether the MDC of 10 degrees applies to all joints studied in this work.

We edited the sentence to make it more accurate. 

20. Lines 384 – 387: “Careful examination” of the movements with lower inter-rater reliability as claimed in this sentence has not been presented in this manuscript. Further examination to support this hypothesis is required to make this claim. Please reword this sentence if further analysis on this hypothesis is not possible.

We reworded this sentence to make it more accurate. 

21. Line 416: “appropriate camera placement” is a key component of the webcam-based method presented in this work. Please provide additional detail as to the height and distance of the camera to the participant, as well as how the videos were framed for each movement.

Following an earlier comment, we now added details of the camera placement in the text. 

22. Line 425: Based on the results in Table 2, it is too strong of a claim to state that there was high inter-rater reliability across all joints. Please rephrase this to acknowledge that inter-rater reliability was low for some joints.

We rephrased this sentence to make it more accurate. 

 

Reviewer #2: Overall Impression:

- This study explored the use of a single-camera approach to obtaining human biomechanics by using a machine-learning solution (Blazepose) based on human pose detection. The reliability of this approach is studied in a sample of 25 adult participants, and the accuracy of the approach is compared to that of a marker-based motion capture system. While I’m excited to see validation studies on such systems, I recommend further revisions to align author’s conclusions with the questions tested in this study.

We thank the reviewer for their kind comments. We hope that our revision helps to align our conclusions with the questions. 

Major issues:

- Recommend caution with claim of cost-effectiveness (and accessibility) as this does not appear to be a factor tested in this paper. Without explicit testing, the cost/accessibility of a device-based Blazepose approach to ROM measurement would seem more than that of a simple goniometer.

We modified our claim to more accurately reflect the scope of this study. 

- There is no limitations section in paper.

We added a Limitations subsection that contains the issues that the reviewers mentioned later. 

- Statistical methods seem to be duplicative. For example, the authors use inter-rater reliability, linear regression and root-mean-square-error to describe the accuracy of Blazepose method versus motion capture method.

We removed the RMSE and the linear regression for each participant to reduce the amount of repetitiveness in the analysis. We did, however, keep the ICC and the per-joint linear regression because, as the first reviewer suggested, the latter provides additional information of the relationship between OptiTrack and BlazePose measurements that otherwise is not reflected in ICC. 

Minor issues:

- Beginning line 34, the abstract would benefit from a concise statement of actual findings regarding reliability and accuracy rather than qualitative statements alone.

We added a description of the actual findings. 

- Line 70, authors state that technical proficiency is a barrier to motion capture. Would there not be some technical barriers to implementing the Blazepose solution? While hopefully less barriers, consider acknowledging (in a limitations section) that some technical hurdles still remain for this method.

We presented the webcam-based ROM evaluation method as a proof of concept, which, as the reviewer pointed out, does impose technical barriers. Given the results in this study, we plan to adapt this method for a web-based application in which the users can simply perform ROM evaluation using a graphical user interface. Because of its online nature, users can access the tool via a URL on a compatible digital device with a webcam. We believe that this would help the users to overcome the technical hurdles that they potentially face. We discussed this in the newly added Limitations subsection. 

- Typo on line 84, “have been used” instead of “has been used”

We fixed the typo. 

- Typo on line 87, “examines” instead of examine

We edited the sentence.

- Line 94, while motion capture is an acceptable comparator, the authors have stated the goniometer is one of the most common ways to measure ROM. I would then expect to see this as a comparator in the study.

- Line 95, the last sentence of section claims Blazepose as an alternative to traditional ROM evaluation. Are authors referring to the goniometer as traditional method? How have the authors tested the accessibility of Blazepose versus a goniometer? This sentence would benefit from revision, and contents of sentence would be better suited in results section.

The absence of a reliability comparison between BlazePose and goniometer measurements in our study was primarily due to the unavailability of a certified and experienced practitioner to conduct the goniometer-based measurements. During the pilot phase, we attempted to perform reliable ROM angle measurements using the goniometer ourselves. However, even when conducted by the same researcher on different days, measurements taken by the two authors (XMW and ZQ) and a student in the lab varied by over 20°. This lack of consistency raised concerns about the accuracy and reliability of the results, leading us to exclude goniometer measurements from the experimental protocol.

To address this limitation and ensure a more comprehensive evaluation in the future, we aim to collaborate with an experienced practitioner who can proficiently conduct goniometer-based measurements. By replicating and extending the current study's findings in partnership with such a professional, we can fill the gaps and provide a more robust comparison between BlazePose and goniometer measurements. This limitation has been discussed in the added subsection on Limitations for transparency and to guide future research endeavors.

- Line 103, the methods section would benefit from more details on participant demographics. Were all participants healthy / neurotypical? Were any neurodivergent populations included? Were any amputees included?

We disambiguated the participant demographics. 

- Typo on line 127, “were used” instead of “was used”

We fixed the typo.

- Line 130, if webcam video of participants, was face occlusion used to prevent identification? Based on Figure 1, the red dots provide some occlusion but I would be concerned that the remainder of visible face could potentially be recognized and person identified.

We thank the reviewer raising a concern on confidentiality. We did not record or store the video feed during the experiment. Figure 1 (middle) showed a colleague, who provided her consent to be featured, during the piloting phase. We clarified this issue in the text. 

- Line 143, please clarify whether the ROM evaluation movements are based on a standard or on some other citable reference. Also, it might benefit the audience to explain/justify why more distal joint movements were excluded, e.g. wrist and ankle ROM.

We added a reference to the ROM evaluation chart used in this study. 

Wrist and ankle movements were excluded from the current study because BlazePose’s estimations of the hand and foot position are unreliable and inaccurate. This detail was added to the same paragraph and mentioned in the limitation section. 

- Line 152, the standing and repositioning required for the webcam could be quite challenging for some participants, especially individuals who are physically frail or neurodivergent. These populations often require therapy interventions to address function/quality of life. While I’m excited to see these pose detection systems and expect future iterations will be more robust to human movement in-the-wild, the current system described does have limitations which are important for clinicians to understand.

We thank the reviewer for pointing out the potential challenge while translating research findings to practical use. We noted this point in the added Limitations section. 

- Line 157, for repeated measures of each movement, were these measures all performed during the same session? If so, was there a prescribed time period between measures?

These measures were performed consecutively during the same session. The sentence was revised to reflect this. 

- Line 173, please provide some explanation or reference regarding the choice of pre-processing filter designs. Would cutoff frequency of 5Hz risk filtering out the frequency of some human movements?

We thank the reviewer for inquiring the rationale behind the cutoff frequency setting. To address the reviewer’s question, we looked through the literature and learned that, in fact, the selection of cutoff frequency is more nuanced (Harry et al, 2022; Schreven, Beek, & Smeets, 2015). Given BlazePose’s sampling frequency of 15 Hz, using a cutoff frequency of 5 Hz for smoothing may artificially perturb the signal, resulting in inaccurate filtered position data. Furthermore, because the current study only focuses on identifying a single point along the entire trajectory, smoothing the entire movement trajectory may be unnecessary. Therefore, we decided to remove the data smoothing during preprocessing. Rationale for omitting this step was also added to the manuscript. 

- Beginning at Line 198, the methods to calculate ROM seem overly complex and the justification is confusing. Using motion capture or the Blazepose method, I would expect these to yield joint locations in 3D, which could then be used to define vectors (e.g. for elbow angle, a vector from shoulder-to-elbow would exist and a vector from elbow-to-wrist would exist). The authors could then determine ROM by calculating the angle between vectors at every instance in time (as the authors later describe). While the pivot joint is rotating and translating with time as the authors depict in Figure 2, the ability to determine the angle at each instance in time would make this rotation/translation seemingly a non-factor. Please clarify further (or highlight references) to support using best-fit planes and joints centroids to calculate the ROM.

We thank the reviewer for pointing out the issues related to the ROM calculation method. We originally developed this method because of the instability of the derived angles as the vectors move in 3D space. However, as we reviewed the analysis to create figures to demonstrate this issue, the instability was no longer present! Therefore, without a strong rationale of using the original method, we re-analyzed the data using vectors without the additional projections. We updated the Methods section (2.4) accordingly and the results. 

- Line 327, the authors report that compression around the acromion caused the Optitrak shoulder location to be perturbed. Given that the 3D coordinate data is available, please quantify the perturbation for the reader. Based on Figure 3a, this perturbation visually appears quite small, and its contribution to the higher interrater reliability in the shoulder flexion ROM would also seem small. Likewise, as 3D coordinate data is available, please quantify the “lack of perturbation” of the shoulder marker during shoulder extension movement.

With the updated analysis, we removed this interpretation and described an alternative interpretation to account for the difference between BlazePose and OptiTrack ROM measurement. 

- Line 412, the authors state the webcam approach is affordable and accessible, but these must be interpreted relatively. Do the authors suggest webcam is more affordable/accessible than goniometer? Or motion capture? Furthermore, as affordability/accessibility were not tested in the described study, I suggest these statements be revised.

We revised the sentence to specify the comparison. We also added another sentence in the same paragraph to delineate why this webcam tool could potentially be more accessible than a goniometer.

---

## [Decision Letter · Decision Letter 1]

4 Oct 2023

PONE-D-23-15143R1A webcam-based machine learning approach for three-dimensional range of motion evaluationPLOS ONE

Dear Dr. Wang,

Thank you for submitting your manuscript to PLOS ONE. After careful consideration, we feel that it has merit but does not fully meet PLOS ONE’s publication criteria as it currently stands. Therefore, we invite you to submit a revised version of the manuscript that addresses the points raised during the review process.

Please provide the minor revisions as suggested by Reviewer 1.==============================

We look forward to receiving your revised manuscript.

Kind regards,

Ryan T. Roemmich

Academic Editor

PLOS ONE

Journal Requirements:

Additional Editor Comments (if provided):

Please make the following revision suggested by Reviewer 1: The title and text of figure 5 say "shoulder flexion", but the image and in-text discussion is about elbow flexion. Please fix.

Reviewers' comments:

Reviewer's Responses to Questions

**Comments to the Author**

1. If the authors have adequately addressed your comments raised in a previous round of review and you feel that this manuscript is now acceptable for publication, you may indicate that here to bypass the “Comments to the Author” section, enter your conflict of interest statement in the “Confidential to Editor” section, and submit your "Accept" recommendation.

Reviewer #1: All comments have been addressed

2. Is the manuscript technically sound, and do the data support the conclusions?

Reviewer #1: Yes

3. Has the statistical analysis been performed appropriately and rigorously? 

Reviewer #1: N/A

4. Have the authors made all data underlying the findings in their manuscript fully available?

Reviewer #1: No

5. Is the manuscript presented in an intelligible fashion and written in standard English?

Reviewer #1: Yes

6. Review Comments to the Author

Reviewer #1: the title and text of figure 5 say "shoulder flexion", but the image and in-text discussion is about elbow flexion. Please fix this before publication.

7. PLOS authors have the option to publish the peer review history of their article (what does this mean?). If published, this will include your full peer review and any attached files.

Reviewer #1: No

---

## [Author Response · Author response to Decision Letter 1]

5 Oct 2023

We thank the reviewers for helping us to catch the typo in Figure 5’s caption. We changed the movement from “shoulder flexion” to “elbow flexion”, with the edits marked in green highlight.

---

## [Editor Report · Decision Letter 2]

9 Oct 2023

A webcam-based machine learning approach for three-dimensional range of motion evaluation

PONE-D-23-15143R2

Dear Dr. Wang,

We’re pleased to inform you that your manuscript has been judged scientifically suitable for publication and will be formally accepted for publication once it meets all outstanding technical requirements.

Kind regards,

Ryan T. Roemmich

Academic Editor

PLOS ONE
---

## [Editor Report · Acceptance letter]

12 Oct 2023

PONE-D-23-15143R2 

A webcam-based machine learning approach for three-dimensional range of motion evaluation 

Dear Dr. Wang:

I'm pleased to inform you that your manuscript has been deemed suitable for publication in PLOS ONE. Congratulations! Your manuscript is now with our production department. 

Kind regards, 

on behalf of

Dr. Ryan T. Roemmich 

Academic Editor

PLOS ONE